# Deep time evolution of the Latitudinal Diversity Gradient: Insights from mechanistic models

**Manon Lorcery**[1‡*], **Laurent Husson**[1], **Tristan Salles**[2], **Sébastien Lavergne**[3], **Oskar Hagen**[4,5], **Alexander Skeels**[6]

**1** ISTerre, CNRS, IRD, Université Grenoble-Alpes, Grenoble, France, **2** School of Geosciences, The University of Sydney, Sydney, New South Wales, Australia, **3** LECA, CNRS, Université Grenoble-Alpes, Grenoble, France, **4** Center for Critical Computational Studies (C³S) Goethe-Universität Frankfurt, Frankfurt am Main, Germany, **5** Senckenberg Biodiversity and Climate Research Centre (SBiK-F), Senckenberg Gesellschaft für Naturforschung (SGN), Leibniz Association, Frankfurt am Main, Germany, **6** College of Science, Australian National University, Canberra, Australian Capital Territory, Australia

‡ Manon Lorcery, Current address: ISTerre, CNRS, IRD, Université Grenoble-Alpes, Grenoble, France
* manon.lorcery@univ-grenoble-alpes.fr

**Citation:** Lorcery M, Husson L, Salles T, Lavergne S, Hagen O, Skeels A 2025 Deep time evolution of the Latitudinal Diversity Gradient: Insights from mechanistic models. PLoS One 20(9): e0332766. https://doi.org/10.1371/journal.pone.0332766

**Data availability statement:** All scripts files are available from the https://github.com/manonlrcy/gen3sis_LDG_dynamics_and_drivers/.

## Abstract

The latitudinal diversity gradient (LDG) designates the increase in species richness toward the tropics. While geological and climatic changes are recognized as key drivers, the precise factors and their relative contributions to species richness gradients remain debated. Using a spatially explicit eco-evolutionary model, we simulate diversification over 125 million years. We validate the model with empirical mammalian richness patterns, and uncover a pivotal role of paleoclimate and paleogeography. This approach allows us to investigate both the mechanisms driving the LDG and space and time variations in species diversification rates across dynamic landscapes integrating changes in tectonic, climatic and surface processes. We show how scale-dependent surface processes are a key driver of regional diversity patterns and how LDG can emerge under a wide range of eco-evolutionary scenarios. Plate tectonics and the subsequent enduring uneven distribution of land masses within the North and South hemispheres imprinted an asymmetric pattern of species diversification rates, primarily shaped by paleoclimate and paleogeography and only to a lesser extent by surface processes. Our simulations also indicate that the LDG has persisted since the Cretaceous, steepened and stabilized from the early Cenozoic on. The modeled scenarios depict that species primarily originate in the tropics and disperse toward the poles without losing their tropical presence. The tropics not only served as a cradle, fostering the origination of new species, but also as a museum, preserving biodiversity over deep time.

## Introduction

One of the most widespread patterns in biological diversity is the decrease in species richness from the low-latitude tropics to high-latitude temperate and polar regions [1]. This spatial pattern of diversity has been observed for most clades [2] and is known as the latitudinal

**Funding:** Funding was provided by the Université Grenoble Alpes under grand IRGA (Tectonic reshaping of the biosphere). OH was supported by the Swiss National Science Foundation (SNSF) P500P B_206815. The funders had no role in study design, data collection and analysis, decision to publish, or preparation of the manuscript.

**Competing interests:** The authors have declared that no competing interests exist.

diversity gradient (LDG). Although first recognized over 200 years ago by naturalists such as Von Humbolt and Darwin, we lack an unequivocal mechanistic explanation [3]. A wide range of hypotheses point towards a handful of potential biotic and abiotic mechanisms, differing in the relative importance they assign to ecological, evolutionary and temporal processes [2,4,5]. Yet, most of these explanations focus on how variations in speciation, extinction, and dispersal manifest themselves across landscapes over time. Despite this shared foundation, our understanding of how paleoenvironmental changes—such as shifts in climate, tectonics, and surface processes—influence these processes to shape biodiversity gradients remains limited. Recent paleoenvironmental reconstructions [6–8] and models of biological diversification on dynamic landscapes [9] now provide a unique opportunity to investigate how Earth system dynamics have shaped the LDG.

Exploring the LDG in deeper geological times may help to unveil the mechanisms at play. However, on an empirical basis, the fossil record and time-calibrated phylogenies yield contrasting insights: some studies show that the shape of the present-day LDG, which shows a sharp tropical peak and poleward decline, has not been a constant pattern throughout the Phanerozoic [5,10]. A present-day-like LDG has been recognized during intervals of the Paleozoic and may have formed, or at least steepened, in the last 30 to 40 My, following a global shift to coolhouse climates at the Eocene-Oligocene Transition, in both the terrestrial and marine realms [11]. As such, present-day-like LDGs are thought to be mostly associated with coolhouse and icehouse periods, while bimodal or even reversed LDGs, with diversity peaks at mid to high latitudes, have been observed during greenhouse climatic periods [11]. More specifically, inferences of past LDGs from the terrestrial fossil record are sparse, but flatter LDGs for some groups have been identified at least during greenhouse periods in the early Paleocene [12] and early Eocene [13]. In the marine realm, on the other hand, present-day-like LDGs are identified as far back as 252 million years ago [11].

A major challenge in studying the history of LDGs is the paucity of fossil data, particularly in the terrestrial realm, driven by distributional and taphonomic biases. Numerical mechanistic modeling permits to circumvent these issues, showing that even simple models using only abiotic factors and static niches successfully predict LDG patterns [14], stable climates and limited dispersal under latitude-dependent environmental conditions can promote tropical richness [14], or that realistic LDGs emerge when eco-evolutionary models account for species' evolutionary responses to dynamic environmental conditions [9,15]. These studies illustrate how mechanistic models help disentangle the relative contributions of ecological, evolutionary, and environmental drivers in shaping the LDG through time.

Since Pianka (1966) [2], LDG studies have primarily focused on climate, evolutionary dynamics, and biotic interactions, leaving the role of physiographic diversity—here referring to variations in surface processes such as hydrology, slope, and terrain—largely unexplored (although the importance of landscape heterogeneity in structuring biodiversity has been identified [16,17]). Here, we explicitly integrate physiographic diversity into Gen3sis, an eco-evolutionary model that simulates populations and species dynamics at the grid-cell level, across geographic landscapes over deep time [9]. It accounts for speciation, extinction, dispersal, and environmental filtering, enabling the emergence of biodiversity patterns, and thus permitting to unravel the intricate links between species diversification processes and environmental dynamics—including surface processes. We compare model predictions to empirical patterns of species diversity, focusing on terrestrial mammals, owing to their extensive and well-documented geographic distributions, and phylogenetic relationships [18]. Our study extends over the past 125 Ma, which enables us to observe biological mechanisms over a geologic time scale that is difficult to achieve solely with empirical data (fossil and phylogenetic).

## Methods

### Mechanistic landscape evolution and eco-evolutionary modeling

Gen3sis [9] is a spatially explicit, population-based mechanistic eco-evolutionary model [19] that integrates detailed biological mechanisms and species interactions to simulate dynamic feedback loops between ecology and evolution. Gen3sis is governed by mechanistic behavior laws that are explicitly designed to predict biodiversity patterns and evolutionary trajectories over time. The model requires i. inputs for the description of the time varying physical environment, that set the boundary conditions (e.g., topography, temperature, precipitation, and land-sea distribution) and ii. parametrization biological functions, or behavior laws (e.g., dispersal ability, speciation, trait evolution, and environmental filtering). For the physical environment, we used a set of global 2° paleo-environmental variables, including reconstructed precipitation ($P$, m/yr) [7], temperature ($T$, °C) [20], physiographic diversity index ($\Phi$, dimensionless) and hydrological categories ($H$, dimensionless) [8] (Fig 1). Physiographic diversity refers to the variation in Earth's surface physical characteristics, including topography, slope, relief and hydrological categories (which encompass different types of water systems, e.g., lakes, rivers). While subsurface processes (mantle convection and crustal deformation) are not explicitly implemented within gen3sis, they drive plate tectonic movements, which in turn shape surface topography and hydrological patterns that are accounted for, over geological timescales. Precipitation, but also lakes and rivers control diversification processes [21]. By incorporating available water resources, we offer a more realistic view of species distribution, where not only mountain building serves as a species pump, but basins and drainage systems also play a role.

Gen3sis runs forward-in-time simulation, meaning it starts with ancestral species and follows their dispersal and diversification across the landscape in discrete time-steps. The model tracks species distributions, traits, and phylogenies at each time step, recording speciation and extinction rates. Here, we simulate diversification of terrestrial mammals over deep time, for which we designed four scenarios–the first scenario ($M0$) is solely based on climate and tectonics, with dispersal and speciation based on geographic distance ($\Delta$). Second scenario $M1s$ integrates physical barriers ($\Phi$) into speciation, with dispersal based on geographic distances ($\Delta$). Dispersal in $M1d$ is based on both geographic distances and physical barriers ($\Delta + \Phi$), with speciation depending solely on geographic distances ($\Delta$). Finally, $M1e$ is drawn from ecological constraints based on environmental suitability (i) and carrying capacity (ii) which includes surface processes ($\Phi$). In all scenarios, isolated populations evolve independently

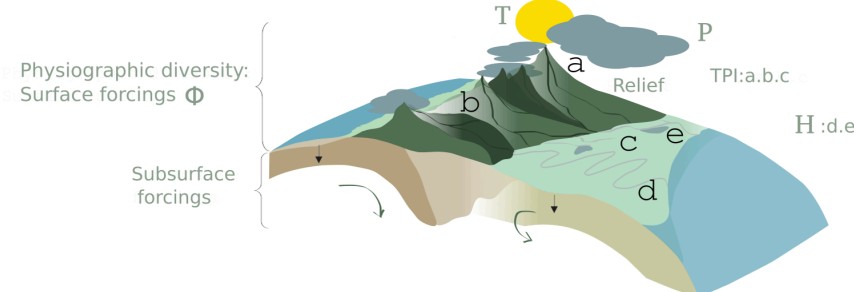

**Fig 1. Paleo-environmental variables used in simulations.** Topographic position index (TPI): a. Mountain tops, b. V-shaped valleys and c. broad flat areas. Hydrological categories (H): d. meandering river, e. lake. P, T and $\Phi$ respectively stand for precipitation, temperature and physiographic diversity.

based on thermal tolerance and speciate upon reaching a divergence threshold. This approach allows us to disentangle how landscape structure, barriers, and ecological factors influence speciation, extinction, and $\alpha$ species richness.

## Landscape model

Our dynamic landscape was generated over the past 150 Ma using reconstructions of the paleo-environnments using a set of global variables. We opted for a 2° × 2° grid, as a compromise between the resolution of climate reconstructions and computational efficiency (see *Acknowledgments*). Paleo-temperatures are based on HadleyCM3L simulations [7] that have been modified to better agree with geochemical proxy data ($\delta18O$) and more equable pole-to-equator temperature gradients deduced from lithological indicators of climate [20]. Physiographic diversity index and hydrological categories were computed from paleo-landscape reconstructions, following our earlier methodology [22]).

More specifically, precipitation, drainage basin index, water discharge and paleo-elevation were interpolated from the landscape evolution model icosahedral mesh on a regular 0.05° grid sourced from goSPL [8]) (Global Scalable Paleo Landscape Evolution [8,23]) which consistently relies on paleo-elevation reconstructions from Scotese & Wright (2018) [6] (PALEOMAP Project) and precipitation grids from Valdes et al. (2021) [7].

We estimate physiographic diversity based on the landscape's structural complexity, derived from topographic reconstructions. One key measure is the Topographic Position Index on each cell $i$ ($TPI_i$):

$$TPI_i = z_i - \sum_{k=1}^{n} \frac{z_k}{n}$$

$$TPI_{S_i} = 100 \cdot \frac{(TPI_i - \overline{TPI})}{\sigma_{TPI}}$$

(1)

that quantifies the difference between local elevation $z_i$ and the mean elevation of its $n$ surrounding cells ($z_k$) within an annular neighbourhood. Because elevation patterns vary with scale, we compute $TPI$ at two spatial resolutions: a finer scale (0.05°–0.15°) and a coarser scale (0.25°–0.5°), and since $TPI$ increase with scale due to spatial autocorrelation in elevation, comparing these raw values across scales can be misleading. To overcome this issue, we calculate standardized $TPI_{S_i}$, in which $\overline{TPI}$ is the mean over the entire grid and $\sigma_{TPI}$ is its standard deviation [22]. $TPI_S$ allow consistent comparison of topographic complexity across scales. We retain 3 morphometric characteristics for physiographic diversity complexity, namely $TPI_S$, and slopes and water fluxes computed from paleo-elevations and precipitations for each time slice. From these continuous variables, we derive categorical variables by defining 10 categories for $TPI_S$, 10 for slope, and 5 for water flux (Table 1).

**Table 1. Hierarchical classification values, parametrisation for slope and water discharge.**

| Category | | | | | | | | | | |
|---|---|---|---|---|---|---|---|---|---|---|
| | 1 | 2 | 3 | 4 | 5 | 6 | 7 | 8 | 9 | 10 |
| Slope (degrees) | <0.03 | ≥ 0.03 | ≥ 0.11 | ≥ 0.3 | ≥ 0.5 | ≥0.85 | ≥ 1.65 | ≥ 2.4 | ≥ 3.5 | ≥ 4.5 |
| Water discharge (Log-scale m³/yr) | <7 | 7 < x < 8 | 8 < x < 9 | 9 < x < 10 | ≥ 10 | / | / | / | / | / |

From these morphometric variables, hydrological categorization H was implemented in the landscape input, based on water fluxes ($km^3/yr$) obtained from simulated landscape evolution [23] (one limitation of the lake data in this dataset is that it does not account for evaporation). From the logarithmic distribution of the water flux, we defined 5 categories (Table 1). From these categorical variables, we calculated physiographic diversity index (Φ) using Shannon's equitability, which is calculated by normalizing the Shannon-Weaver diversity index $d_{SW}$ [8]:

$$d_{SW} = -\sum_{k=1}^{S} p_k ln(p_k)$$
$$\Phi = d_{SW}/ln(C) \tag{2}$$

with $p_k$ the proportion of observations of type $k$ in each neighbourhood, $C$ the number of categorical variables (here $C = 3$), for TPI, slope and water fluxes.

## Model specifics and parameters

Each simulation tracks a clade's radiation from its initial species distribution across reconstructed paleo-environments, incorporating four key processes: dispersal, speciation, evolution, and ecology (Table 2).

**Dispersal.** In gen3sis, the cost function determines the difficulty for species for moving between sites at each time step (here 1 Ma) across the landscape. This cost reflects geographic or environmental barriers that can constrain movement. In our models (Table 2), the default cost function is computed solely on geographical distance. On all models ($M0$, $M1s$, $M1d$, $M1e$), only terrestrial regions are habitable: moving across land induce a baseline cost, while crossing water doubles the cost of dispersal, reflecting the challenge of traversing hostile environments. For model $M1d$, the cost function also accounts for spatial differences in physiographic diversity, meaning that sites with different surface characteristics are harder to reach, reducing connectivity between environmentally dissimilar regions. For all models, at each time step, each local population $i$ can disperse into other sites $s$ from a dispersal kernel drawn from a continuous probability distribution (Weibull, centered on a 2° grid), allowing dispersal beyond immediately neighboring sites, with shape $\phi = [2, 3]$ and scale $\Psi = [100, 600]$ (Table 3). This results in most dispersal values being around 250 to 1500 km, with rare large dispersal events above 1750 km.

Table 2. **Model forcing framework for testing LDG drivers.** Population dispersal is determined either by geographic distance alone, $\Delta$ ($M0$, $M1s$, $M1e$) or by a combination of geographic distance and physical barriers—such as mountains and rivers—$\Delta + \Phi$) ($M1d$). Speciation begins when populations of a species become isolated either: by geographic distances $\Delta$ ($M0$, $M1d$, $M1e$) or by physical barriers ($\Phi$) ($M1s$). These isolated populations evolve independently through time based on their thermal tolerance ($M0$, $M1s$, $M1d$, $M1e$). Diverging populations become distinct after reaching a threshold of differentiation. Species ecology is drawn from environmental suitability based on species' niche (i) and carrying capacity (ii).

| Model | Configuration | | | |
|---|---|---|---|---|
| | *Dispersal* | *Speciation* | *Evolution* | *Ecology* |
| M0 | $\Delta$ | $\Delta$ | Thermal tolerance evolving randomly | (i) $T$ and $P$, (ii) $\alpha = 0.5$; $\beta = 0.5$; $\gamma = 0$ |
| M1s | $\Delta$ | $\Phi$ | | (i) $T$ and $P$, (ii) $\alpha = 0.5$; $\beta = 0.5$; $\gamma = 0$ |
| M1d | $\Delta + \Phi$ | $\Delta$ | | (i) $T$ and $P$, (ii) $\alpha = 0.5$; $\beta = 0.5$; $\gamma = 0$ |
| M1e | $\Delta$ | $\Delta$ | | (i) $T$ and $P$, (ii) $\alpha = 0.33$; $\beta = 0.33$; $\gamma = 0.33$ |

$\Delta$: geographic distances; $\Phi$: physiographic diversity; H: hydrological categories; $T$: species thermal niche; $P$: species precipitation niche.

**Speciation.** Speciation follows an allopatric model, where geographically isolated populations undergo genetic divergence at each time step. Once the cumulative divergence crosses a speciation threshold $\tau$ it is considered a new species and evolves independently (Bateson–Dobzhansky–Muller model of genetic incompatibility [24]). Speciation occurs after $\tau =$ [0.5, 3] (Table 3), corresponding to events occurring after 0.5 to 3 Ma of isolation in cases with a constant diverging rate, which is based on estimated times for reproductive isolation to establish [25]. While this simplified (for computational cost) framework captures key aspects of geographic speciation, it may influence the results—for example by potentially overemphasizing the role of spatial isolation.

**Evolution.** Trait values follow a normally distributed stochastic process directed by environmental temperature, such that evolutionary changes tend to align with local conditions while also incorporating random variation that reflects natural evolutionary uncertainty. Divergence rates between populations follow a normally distributed stochastic process (Brownian motion) directed by environmental conditions, reflecting how factors such as geographic isolation influence the pace and direction of trait evolution. At each time step, the evolutionary change in the temperature niche $\overline{Ti}$ for a given species is modelled as follows: $\Delta\overline{Ti} = |\mathcal{N}(0, \sigma^2)|$, where $\Delta\overline{Ti}$ is the fluctuation around the local average temperature $\overline{Ts}$, where $\Delta\overline{Ti}$ represents the change in temperature niche, and $\sigma = 0.005$ (Table 3) represents the standard deviation governing the magnitude of random change, corresponding to $\pm0.5°C$ per time step. It follows the reconstruction of trait values of ancestral mammal [26] and vertebrate species [27]. The updated temperature niche $\overline{Ti_n}$ is drawn from:

$$\overline{Ti_n} = \overline{Ts} - \left(\overline{Ti} + \Delta\overline{Ti}\frac{\overline{Ts} - \overline{Ti}}{|\overline{Ts} - \overline{Ti}|}\right) \tag{3}$$

**Ecology.** The size $N$ of population $i$ in site $s$ varies depending on species' environmental suitability $K$ and carrying capacity $K_s$. $K$ is given by a Gaussian function of the thermal and precipitation niche, which declines with increasing distance between the local temperature and precipitation values ($\bar{T}_s$ and $\bar{P}_s$) and the species' temperature and precipitation optima ($\bar{T}_i$ and $\bar{P}_i$):

$$K = K_s \cdot exp^{-\left(\frac{\bar{T}_i - \bar{T}_s}{\omega_t}\right)^2} \cdot exp^{-\left(\frac{\bar{P}_i - \bar{P}_s}{\omega_p}\right)^2} \tag{4}$$

where $\omega_t$ and $\omega_p$ determine the strength of environmental filtering (Table 3). Niche width evolve in all models with $\omega_t$ and $\omega_p$ = [0.05, 0.25] (corresponding to niche widths of $\sim 1.6$ to $\sim 8.25°C$ for $\omega_t$ and $\sim 0.7$ to $\sim 1.6$ m/yr for $\omega_p$, Table 3). The carrying capacity $K_s$ for each

**Table 3. Model parameter ranges.**

| Stochastic Ranges | Range values | |
|---|---|---|
| | *min* | *max* |
| Threshold for divergence (Myr) ($\tau$) | 0.5 | 3 |
| Temperature niche width ($\omega_t$) | 0.05 | 0.25 |
| Precipitation niche width ($\omega_p$) | 0.05 | 0.25 |
| Dispersal shape ($\phi$) | 2 | 3 |
| Dispersal scale ($\Psi$) | 100 | 600 |
| **Fixed parameter** | **Value** | |
| Evolutionary parameter ($\sigma$) | 0.005 | |

2° × 2° cell is set to:

$$K_s = min(K_{max}, (\alpha \cdot \bar{P} + \beta \cdot \bar{H} + \gamma \cdot \bar{\Phi}) \cdot K_{max}) \cdot cos(lat) \tag{5}$$

where $cos(lat)$ accounts for the latitudinal variation of cell areas, $K_{max}$ is the maximum carrying capacity (fixed to 1 in all models), $\bar{P}$, $\bar{H}$ and $\bar{\Phi}$ are respectively the precipitation rates, hydrological categories and physiographic diversity index (normalized by their maximum values) and $\alpha$, $\beta$ and $\gamma$ the scaling factors for each variables in available resources balance (Table 2). Only model $M1e$ accounts for physiographic diversity ($\gamma > 0$). The change in population size in each site per timestep is expressed as follows:

$$dN = N \cdot (K - N) \tag{6}$$

where $N$ is the population size and $K$ is the growth potential of the population at each timestep, that depends on $K_s$. Species with broad niches do not automatically reach higher abundances unless their suitability remains high across sites.

When species colonize or become locally extinct (when $N < 0.01$), abundances of all species are reapportioned according to each species' environmental suitability, such that well-adapted species obtain a higher abundance. Complete extinction of a species arises when it no longer occupies any site.

In order to account for the stochasticity of our models, all 4 models described in Table 2 are run over 100 simulations covering ranges of ecological parameters (Table 3) explored using Sobol sequences, a quasi-random number generator that evenly samples parameters in a multidimentional space [28]. To define the parameter ranges – speciation threshold $\tau$, temperature trait optimum $\sigma$, niche width and dispersal boundaries ($\Psi \phi$)—we set upper and lower boundaries based on existing literature [9,25–27,29] and modeling explorations. These simulations were subsequently averaged for each model. Models run forward over 150 Ma, with 1 Ma time steps, enabling efficient simulation over geologic timescales, but admittedly representing a coarse resolution for some ecological and evolutionary processes that can occur over shorter timescales. In the following, we exclude the initial 25 Myr spin-up phase, during which random initial conditions impact the outcomes (based on an analysis of model output variability, where the spin-up phase was characterized by large amplitude fluctuations in $\alpha$ richness between time steps, while the post-spin-up phase showed dynamic but relatively smaller amplitude variations).

The 150 Myrs time frame includes the K-Pg transition; however, it is essential to note that our models are not designed to replicate specific mass extinction events, but primarily focus on broader patterns that limit their relevance for direct comparisons with empirical observations of specific events. Our scenarios represent a *pseudo K-Pg transition*, emerging solely from harsher climatic conditions, allowing us to focus on the general underlying mechanisms.

## Benchmarking and analyzing LDG through time

We first benchmark our models against the present-day empirical patterns of terrestrial mammal richness. Validating the model in this way ensures that its structure and assumptions are sufficient to capture the key features of the modern LDG.

Building on this validation, we then explore how the LDG has evolved throughout geological time. To quantify its spatiotemporal dynamics, we fit a hyperbolic tangent function to modeled richness curves for each hemisphere at every time slice, such as $\alpha(lat) = \alpha_{min} + \Delta_\alpha tanh(\theta (lat - lat_0))$, having $\Delta_\alpha$ the amplitude of the LDG curve, $\theta$ the slope, $lat_0$ the

hyperbolic tangent midpoint, and $\alpha_{min}$ the minimal species richness (Fig 4). We retain slope and width, respectively indicative of the characteristic gradient and width of the LDG.

To further understand the specific contribution of environmental factor on LDG, we compared paleo-environmental parameters—mean annual temperature (MAT), mean annual precipitation (MAP), and elevation—with biodiversity metrics generated by the modeled scenarios. To explore the influence of paleogeography and geological barriers, we calculated the mean land surface above sea level and mean elevation as functions of latitude across geological time. To understand whether palaeoclimate influenced the biogeographical distribution of terrestrial mammals, we concatenated simplified Köppen Climatic Belts [20,30,31]: Tropical regions (A) were defined as areas with $MAT > 18°C$ and $MAP > 0.6$ m/yr; arid regions (B) were identified using a critical precipitation threshold $\kappa = 2 * MAT$; regions with $MAP < \kappa$ were classified as arid. Here, we used a simplification of the Köppen classification original criteria for distinguishing semi-arid from arid climates based on seasonality. Temperate regions (C) account for areas with $5°C < MAT < 18°C$ and $0.5 < MAP < 2$ m/yr, while continental climates (D) were identified by $-5°C < MAT < 10°C$ and $0.4 < MAP < 1$ m/y. Polar climates (E) were defined solely based on $MAT < 10°C$.

## Results

### Present-day mammal diversity

All models generate a highly congruent pattern with empirical data (Spearman correlation > 0.9) and sharp LDG (Fig 2A–2B). However, species richness was underestimated in some regions, particularly in Amazonia and equatorial latitudes between –15° and 15°, likely due to dry biases in precipitation reconstructions [7]. Regionally, $M1e$ tends to predict higher richness than other models $M0$, $M1s$, $M1d$ especially in the Northern hemisphere but also in Amazonia and in the Hengduan region (Fig 2A). This suggest that integrating surface processes into niche ecology enhances diversity in these regions by increasing niche heterogeneity and ecological opportunity.

Abiotic variables present clear latitudinal gradients, with tropical peaks and poleward declines (Fig 2C). However, the high similarity in richness patterns across multiple model scenarios suggests that local-scale surface processes play a limited role in shaping the global-scale gradient. It suggests instead that the LDG more plausibly arises from evolutionary responses to dynamic environmental conditions, including long-term tectonic shifts and climate variability; even with varied ecological settings, the richness gradients remain relatively stable, pointing to deeper evolutionary and macroenvironmental mechanisms as the primary LDG drivers. Moreover, the strong correlations between empirical richness maps and LDG patterns and the mechanistic models confirm the plausibility of the modeled scenarios. Using this present-day congruence as a benchmark allows us to investigate how long-term environmental dynamics have influenced biodiversity patterns through time.

### Deep time

By examining how speciation and extinction patterns respond to paleogeographic and climatic changes, we hypothesize that the current biodiversity is not a state function of environmental variables, but rather depends on its history.

**Paleogeography and biodiversity dynamics.** The clearest way to observe biodiversity over deep time within our models comes from the spatial patterns of predicted species richness, speciation and extinction rates (Fig 3a–3c). While the Cretaceous features broader latitudes of high richness—particularly around 30°N—this pattern progressively narrowed

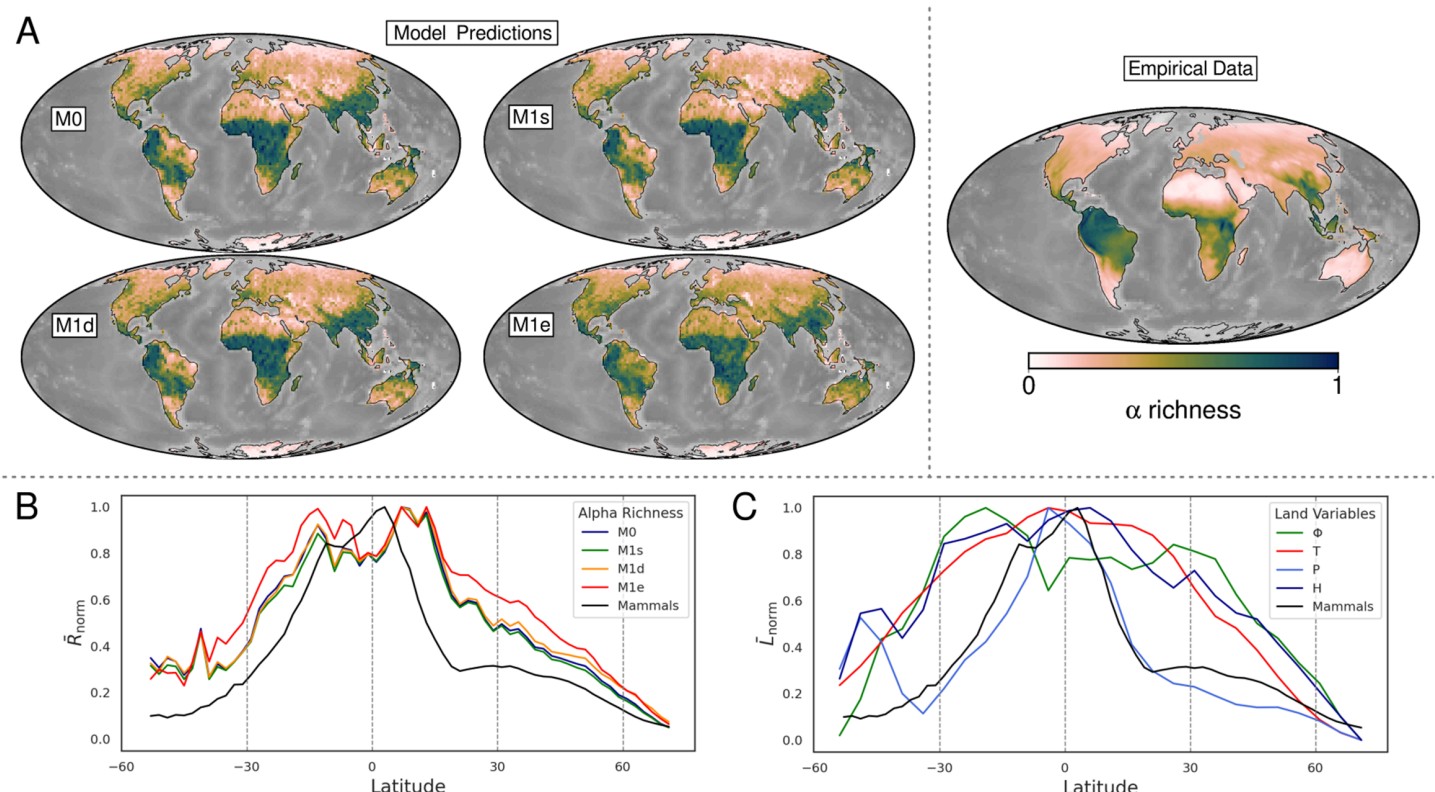

**Fig 2. Present day modeled and empirical terrestrial mammal $\alpha$ richness and the Latitudinal Diversity Gradient (LDG).** A. Model average predicted (left) and empirical (right [32]) richness of terrestrial mammals. Both modeled and empirical richness are normalized to their maximum value. B. Comparison of model results (M0, M1s, M1d, M1e) with empirical data [32] (black curve) on terrestrial mammals showing the present-day LDG left). Diversity is the mean richness normalized to its maximum value, per latitudinal degree. C. Mean environmental input variables, also area-scaled and normalized to their maximum values, per latitudinal degree, where : stands for physiographic diversity, T: temperature, P: precipitation, and H for hydrological categories. To ensure comparability with empirical data, mean values were calculated within the latitudinal range of –54° to 71°. Bathymetry is mapped using PALEOMAP reconstructions [6]. Reprinted from [6] under a CC BY license, with permission from C.R. Scotese, original copyright 2018.

toward the equator throughout the Paleogene and Neogene (Fig 3a). Similarly, speciation rates transitioned from widespread Northern Hemisphere peaks during the Cretaceous to tropical concentration by the Eocene (Fig 3b). Notably, the transient shift in speciation rates following the post-*pseudo K-Pg transition*, reflects a transition from higher latitudes to equatorial regions. Extinction patterns are consistently higher in the Northern than the Southern Hemisphere throughout the Cretaceous and Cenozoic (Fig 3c). This latitudinal imbalance reflects the long continental isolation of Southern landmasses—such as the separation of Australia from Antarctica and the isolation of South America and Africa—causing lesser widespread extinctions in these regions, and restricting dispersal opportunities. Net diversification rates (speciation - extinction) provide a measure of net biodiversity change over time. Strikingly, high rate are remarkably centered in the tropics (Fig 3e), reflecting both high speciation and low extinction rates. Turnover ((speciation + extinction)/$\alpha$ richness), defined as the rate of species replacement due to the combined effects of speciation and extinction, is higher in the Northern hemisphere (Fig 3f), especially at high latitudes with low richness and high extinction rates.

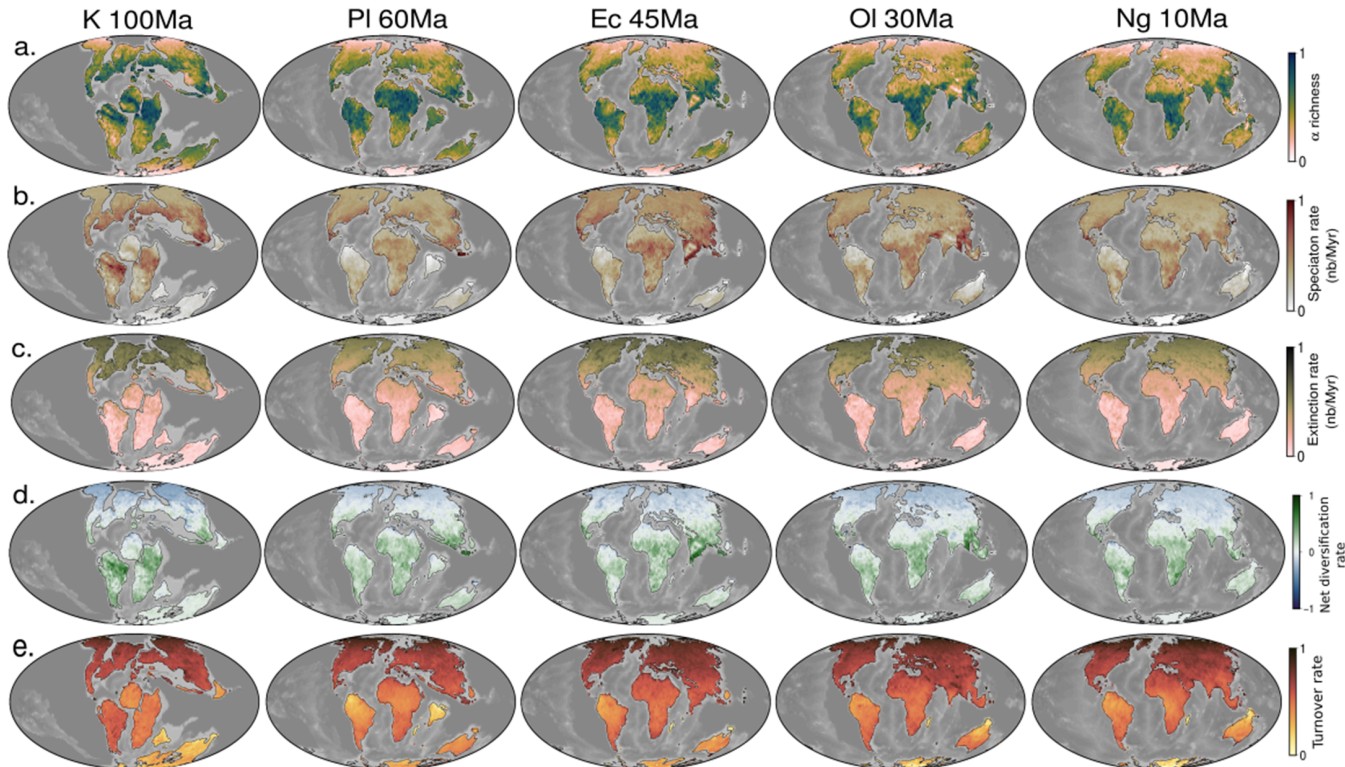

**Fig 3. Paleogeography and main biodiversity drivers from modeled scenarios, model _M1e._** (a) Richness, (b) Speciation, (c) Extinction, (d) Net diversification rate (Speciation − Extinction) and (e) and Turnover ((Speciation + Extinction)/Richness), over deep time, normalized to their maximum mean value. Paleo-diversity maps are represented using average model M1e as an example, given the high similarity in outcomes across models. Bathymetry is mapped using PALEOMAP reconstructions [6]. Reprinted from [6] under a CC BY license, with permission from C.R. Scotese, original copyright 2018.

The narrowing ranges of $\alpha$ richness and speciation to the tropics reflects the combined influence of abiotic factors and ecological opportunities in promoting both high diversity and allopatric speciation. In contrast, the asymmetry in extinction—persistently higher in the Northern Hemisphere—stems from reduced dispersal across hemispheres and the prolonged isolation of southern landmasses. These spatial patterns highlight the paleolatitudinal dependence of diversification, inviting to analyze the interplay of dispersal, ecological and evolutionary mechanisms, and environmental drivers in shaping the LDG.

**LDG dynamics.** The LDG offers a lens to explore how species richness has been distributed across latitudes over time. By reducing the spatial complexity into a latitudinal dimension, we opt for a streamlined approach that we further collapse into a hyperbolic tangent function fitting to LDG curves, for each hemisphere, that provides a quantitative framework to trace the emergence and stability of these gradients.

Both hemispheres reveal three distinct phases (Fig 4): (i) before the K-Pg, the LDG is relatively flat and wide; (ii) during the _pseudo K-Pg transition_, it becomes very steep and narrow; and (iii) after the K-Pg, it gradually steepens and narrows further to the equator. LDG shifts during the _pseudo K-Pg transition_ are driven by an abrupt decline in precipitation and temperatures, which concentrates richness in the tropics. Following the _pseudo K-Pg transition_, the LDG width stabilizes in the Northern Hemisphere, but fluctuates more in the Southern hemisphere. In the Northern Hemisphere, LDG steepening reflects a decline in $\alpha$ richness

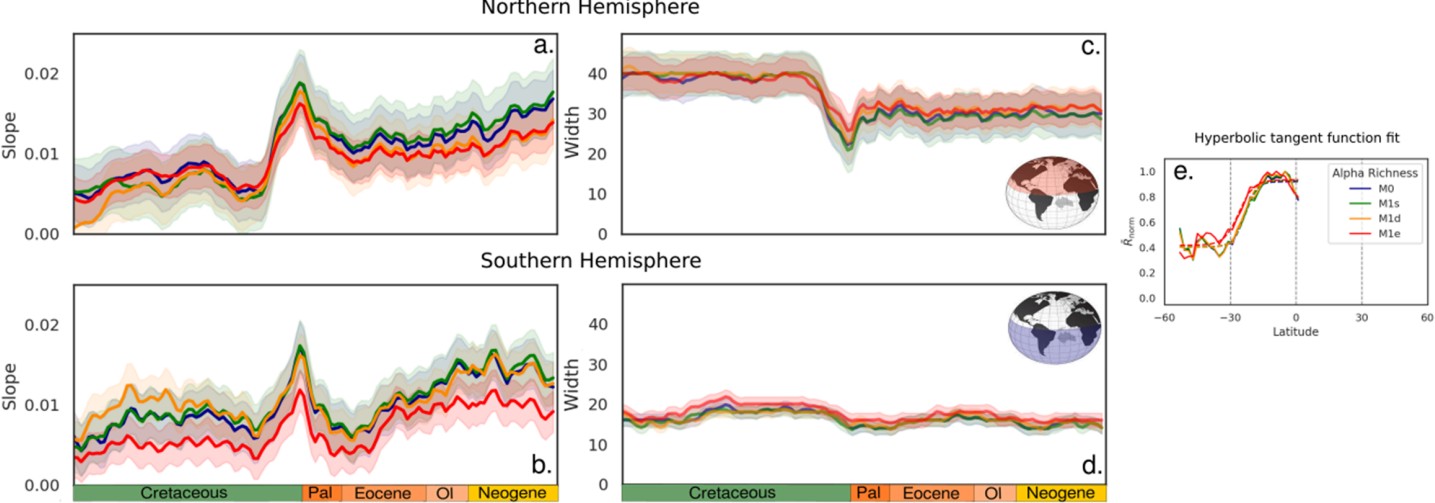

**Fig 4. LDG Dynamics.** LDG slopes (a. and b.) and widths (c. and d.) derived from a hyperbolic tangent function fitted to the latitudinal $\alpha$ diversity curves and estimated separately for the Northern (a. and c.) and Southern (b. and d.) hemispheres using absolute latitude. Example of the hyperbolic tangent function fit for each modeled scenario for the Southern Hemisphere (e). LDG curves correspond to the normalized mean $\alpha$ richness per latitudinal degree, for models M0 (blue), M1s (green), M1d(orange), and M1e (red).

near 30°N (Fig 3a) driven by a shift from mid-latitude landmass concentration in the Cretaceous to a more even tropical-to-polar spread in the Cenozoic. Contrastingly, $\alpha$ richness near 30°S was consistently lower, and the LDG width about half as large, highlighting greater boreal diversity.

These patterns unveil key insights into the spatial distribution of biodiversity over time, revealing distinct differences between the hemispheres and the influence of plate tectonics and climatic shifts. The progressive rearrangement of landmasses not only shaped species distributions but also influenced the strength and structure of LDGs by altering habitat availability and dispersal routes.

**LDG patterns and biodiversification processes.** At first order, the models demonstrate similar outcomes, indicating that fine-scale surface processes—rather than broad evolutionary or biogeographic mechanisms—play a limited role in shaping large-scale diversity gradients. Instead, broad diversification patterns are primarily driven by the common attributes to all models: plate tectonics, which define the land-sea mask, and climate, which set the geographical ranges of habitability. At regional scales, however, models behave differently, reflecting the specific parametrization of each model scenario. This consistency enables a deeper investigation into the fundamental ways paleoenvironmental changes shape biodiversity gradients through the effect on ecological, temporal, and evolutionary factors driving the LDG. Given this similarity (*Supporting information)* we focus in the following on model M1e (Fig 5A). $\alpha$ richness and speciation rates are consistently higher in tropical regions (Fig 5). However during the Early Cretaceous, species richness and speciation rates peak at higher latitudes (∼30°N) for the Northern hemisphere (Fig 5A)—likely triggered by continental fragmentation. Globally, species richness and speciation trends steadily increase from the Early to Late Cretaceous, followed by an abrupt decline at the K-Pg boundary and a subsequent rapid recovery (Fig 5B). Extinctions, similarly to turnover rates (Fig 5A), occur mainly at high-latitudes (particularly in the Northern hemisphere), characterized by greater environmental instability and isolation, where continental connections (essentially with and within

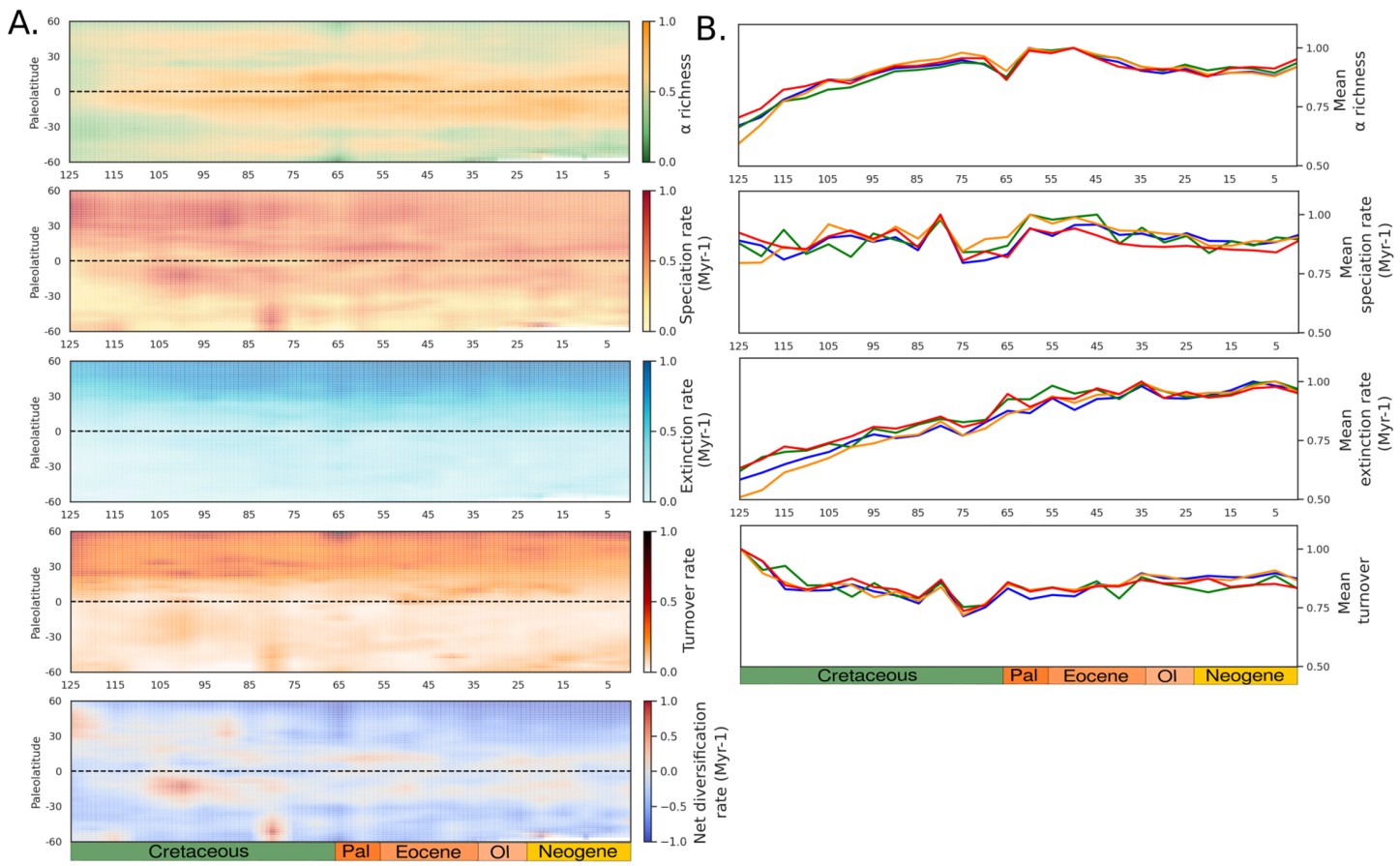

**Fig 5. Modeled mammalian biodiversification across paleolatitudes and at global scale.** A. Paleolatitude figures are represented using model M1e as an example (outcomes from all other models being similar at first order, see Fig S1, Fig S2, Fig S3). Each variable is measured as area-scaled, representing normalized to their maximum mean biodiversity metric per latitudinal degree. B. Normalized to their maximum mean biodiversity metric at global scale. Color-coded models: M0 (blue), M1s (green), M1d (orange), and M1e (red).

Eurasia) were more extensive. This pattern underscores the role of dispersal in facilitating poleward expansions while also driving extinction rates. Furthermore, a significant increase in turnover rates is observed from the Oligocene onwards (Fig 5B). As seen previously, despite different setups, modeled scenarios produce similar trends. However, models incorporating surface processes (*M1s*, *M1d*, *M1e*) diverge in turnover trend during the Cretaceous and post-Oligocene compared to *M0*, which lacks surface dynamics (Fig 5B). Maximum net diversification rates are also found in tropical regions (Fig 5A), showing both constant high net diversification in the tropics and patchy distributions at mid to high latitudes, reflecting bursts of speciation.

Overall, these results highlight the role of biotic processes in shaping the LDG by governing how species originate, expand, and persist across space and time. Dispersal enables range expansion, while also driving poleward extinction patterns; ecological filtering favor species persistence within broader and more stable niches in the tropics, allowing for long term persistence. These results also underscore that speciation itself is not solely driven by geographic isolation—though physical separation, especially at mid-to-high latitudes due to continental fragmentation, can trigger bursts of speciation. More broadly, speciation also arises from

combined ecological opportunities, abiotic factors, and trait evolution, which are particularly prevalent in the tropics. This interplay results in high and stable speciation rates in tropical regions, reinforcing their role as long-term sources of biodiversity. In the following, we further investigate how interactions between biotic and abiotic factors contribute to shaping the LDG.

### Paleo-environments and biodiversity drivers

Our models indicate that the global biodiversity patterns evolve, at first order, regardless of the specific parametrization of each model, which conversely operates at regional scales. These first order behavior also suggests that, beyond continental isolation, mountain building and the exposure of landmasses to climatic conditions also play important roles. To understand their specific contributions, we compare paleo-environmental parameters with modeled biodiversity metrics.

The Northern Hemisphere consistently featured larger and more elevated landmasses than the Southern Hemisphere (Fig 6), providing more space for terrestrial diversification and facilitating species dispersal from the tropics. Between the Cretaceous and Oligocene, land area expanded between 15°N and 30°N due to the gradual closure of the Tethys Sea. These tectonic processes also uplifted the Central Asian Orogenic Belt (CAOB) and the Laramide

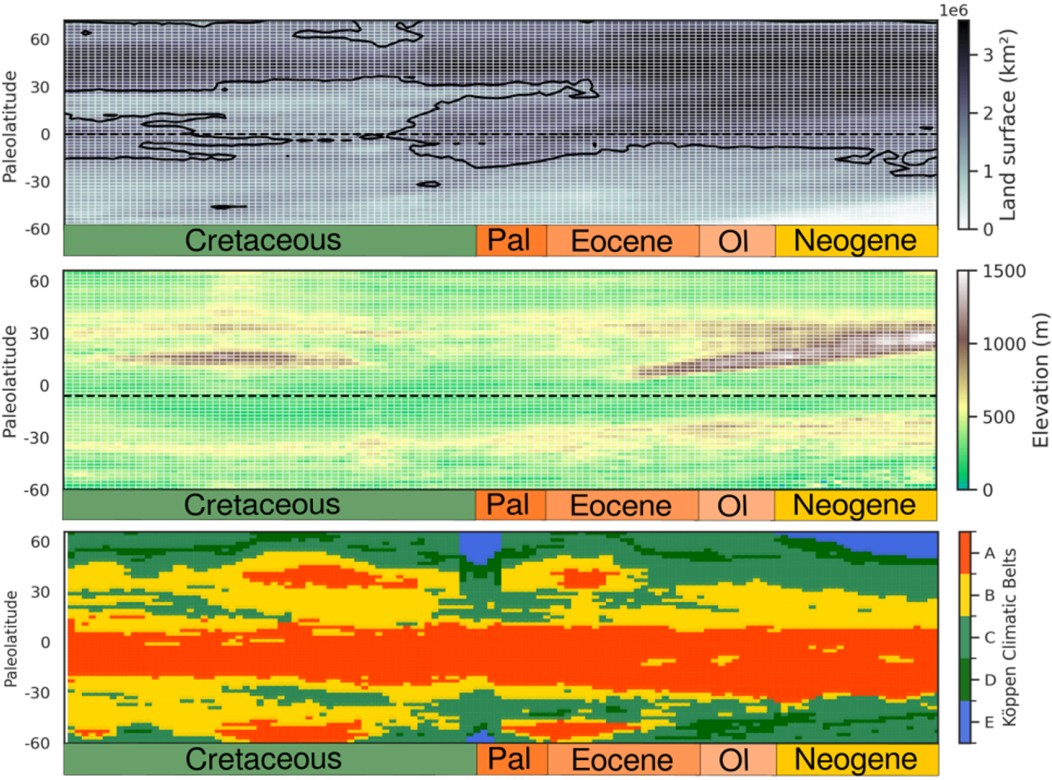

**Fig 6. Landscape dynamics over time.** From top to bottom, each landscape variable is represented as a function of paleolatitude: Mean land surface above sea-level (km²) with isocontour representing the median land surface; Mean elevation above sea-level (m) and simplified Köppen climatic belt, with tropical (A), arid (B), temperate (C), continental (D) and polar (E) regions.

orogeny during the Cretaceous, contributing to the high net diversification observed near 30°N (Fig 5A). From the Eocene onward, elevation in the Northern hemisphere remained high due to the Alpine orogeny (*sensu lato*, from the Pyrenees to the Hengduan mountains). Mountain building further promoted biodiversity by creating geographic barriers, enhancing allopatric speciation.

In addition to changes in land surface and elevation, climate also underwent substantial shifts over time, except in equatorial regions, where tropical climatic stability prevailed (Fig 6). During thermal maxima, such as the Cretaceous Thermal Maximum (85–90 Ma) and the Paleocene-Eocene Thermal Maximum (56 Ma), tropical conditions extended to higher latitudes (boreo-tropical zones, Fig 6). In contrast, the K-Pg transition marked a cooler period, with narrower arid zones, expanded temperate regions, and more extensive polar environments (Fig 6). Throughout the Neogene, climatic regions remained relatively stable (Fig 6).

As observed previously, models indicate that tropical regions consistently harbor the highest species richness from the Cretaceous onward (Fig 5A), while also driving high speciation rates during thermal maxima (Fig 5B), highlighting the importance of warm, stable climates in fostering both the origin and persistence of biodiversity. High $\alpha$ diversity and speciation rates in the tropics likely arise from a combination of climatic stability, extensive land availability, and biotic processes—including ecological opportunity, and trait evolution. In contrast, extinction and turnover are highest at high northern latitudes (Fig 5A), where elevated land and fluctuating climates—shifting between temperate, continental, and tropical—created environmental instability. During the Eocene, the expansion of boreotropical regions, combined with larger Northern Hemisphere landmasses, increase habitat availability, enhancing speciation and $\alpha$ richness. In contrast, the reduction of tropical landmasses during the Late Cretaceous (Fig 6) trigger a surge in speciation, followed by a sharp decline. This pattern highlights how continental fragmentation can drive diversification by isolating populations, fostering speciation processes up to a threshold. Beyond this point, specialization within isolated populations leads to reduced diversity, suggesting that dynamic continental configurations play a more critical role in driving diversification than stable continental plates.

These patterns emphasize the interplay of abiotic and biotic forces in shaping biodiversity: stable climates promote species persistence and divergence through trait evolution and ecological filtering; large landmasses enhance ecological opportunity and facilitate wide dispersal; while the combination of both landscape connectivity and mountainous regions drives divergence through habitat isolation, especially at mid to high latitudes, where climate instability limits diversification.

## Discussion

Our models are agnostic, in the sense that they are not designed to advocate, or even simply test any specific theory. However, because our *in silico* experiments stem from evolutionary scenarios, their predictions are useful to explore the mechanisms underpinning the dynamics of emergent properties like the LDG, as we do in the following discussion.

### Landscape dynamics and their legacy on biodiversity

Our models suggest that physiographic diversity has a limited impact on global LDG patterns, while playing a notable role at regional scales (for instance in the Hengduan region), highlighting a scale-dependent effect: physiographic diversity can shape diversification locally or within specific biogeographic regions, without contributing to broader diversity patterns. For instance, in all models (*M*0, *M*1*s*, *M*1*d*, *M*1*e*), high turnover rates were observed during the

mid-to-late Cretaceous (ca. 90 Ma) and during periods of intense geodynamic activity (orogenesis, plate reorganizations, and expansion of inland seas) [33]. These tectonic shifts triggered physiographic diversity, and were accompanied by major transgression events. Lowland habitats shrank and repeated cycles of transgression and regression forced upland range shifts among lowland populations, while habitat fragmentation enhanced diversification during this period explaining why model *M1e*—which accounts for physiographic diversity as an ecological parameter—displays higher turnover than *M0*.

Furthermore, models that incorporate surface processes (*M1*, *M1s*, and *M1e*) reveal higher turnover rates during the Late Cretaceous. This supports the hypothesis proposed by Weaver et al. [33], who suggested that the drivers of the Cretaceous Terrestrial Revolution (KTR) extend beyond tectonic and climatic shifts to include surface processes that influenced mammalian diversification. Our findings accordingly show that the interplay of tectonics, climate, and surface processes enhance landscape complexity and ecological heterogeneity and promoted speciation, dispersal, and niche diversification. This transient event amplified turnover by generating variable habitats across spatial scales, particularly in mountainous regions and alluvial systems. These results highlight that during transient tectonic and climatic periods, landscape complexity further amplifies biodiversity. Recognizing that mountainous regions foster biodiversity—especially in warmer climates—we propose that the KTR was initiated by intensified tectonic activity and amplified by increasing landscape complexity and climatic shifts [22]. Together with transient geological and climatic events, surface processes contributed to the expansion of ecological opportunities during the KTR.

## From isolation to specialization: Mechanisms of the LDG

Our results reveal high speciation and persistent $\alpha$ richness in the tropics, and elevated extinction rates at higher latitudes, in which the tropics act as both a cradle and museum of biodiversity. This extinction pattern aligns with mammalian phylogenies [34–36] (exceptions exist in some orders, such as Lagomorpha [34], which exhibits inverse LDGs), and fossil records from marine taxa [1,37]. Previous studies have explored the temporal, ecological and evolutionary factors that shape the LDG [2,4,5]. Here, we show that these factors can be better understood in the context of paleoenvironmental changes, and that their interaction across dynamic landscapes is a key factor. Temporal factors, which include the "time-for-speciation" effect, reflecting the long-term persistence of clades in the tropics, are also emerging from our simulations, under a wide range of eco-evolutionary assumptions. Tropical climatic stability since the Cretaceous indeed enabled long-term evolutionary persistence and divergence in the tropics.

Our simulation results also point out that ecological mechanisms—such as the role of area, energy availability, and environmental stability, are also important drivers of the LDG, a result well in line with past the literature about key LDG factors [2,4,5]. While these factors have been widely acknowledged, their individual effects often remain confounded. Here, our models help disentangle the influences of climate and geography. For instance in the Early Cretaceous, elevated net diversification at higher latitudes coincided with the presence of large, contiguous landmasses in the Northern Hemisphere. The long-standing N-S asymmetry in the distribution of land masses enhanced ecological opportunity and connectivity, promoting dispersal and speciation [4] in the Northern hemisphere. As continental fragmentation increased during the Late Cretaceous, the resulting habitat isolation—particularly in tropical inland seas and basins, but also at mid to high latitudes—catalyzed allopatric speciation.

Evolutionary factors emphasize differences in diversification dynamics and dispersal asymmetries across latitudes, including higher net diversification rates in the tropics [29,38]. Our

models further show high extinction and turnover rates at high latitudes are associated with climate instability and ecological filtering, favoring narrow physiological tolerance and specialization, which helps accumulating species in tropical regions. This pattern suggests a complex interplay between environmental stability and evolutionary constraints, highlighting both ecological and evolutionary mechanisms at play. Although the Cretaceous and Eocene both experienced climatic optima, our models show high net diversification rates beyond tropical latitudes during the Cretaceous, which underscores that climate alone cannot explain the LDG. This challenges the assumption that species richness is primarily driven by large-scale climatic shifts [10,11,13], and reinforces the central role of geographic structure, or the 'Geographical Heterogeneity Hypothesis' [11] distinct from the area hypothesis: it is not only the total land area that influences diversity patterns, but also its latitudinal distribution [11].

Last, our models do not explicitly include biotic interactions such as mutualism or competition [39]. In the tropics, more numerous species interactions may lead to niche differentiation, enabling more species to coexist and greater speciation rates [4,40]. While we cannot rule out this possibility, our results show that such interactions are not required to reproduce observed latitudinal diversity patterns. Overall, our findings challenge the idea that climate alone is the primary ecological LDG driver, instead assigning a central role of paleogeography and continental connectivity.

## Historical dynamics of the LDG, origins of present-day biodiversity

The main trend that our models show is that the LDG became established during the Cretaceous, and that the present-day LDG did so at around 50 Ma, then steepening, and taking its current form by approximately 35 Ma. This Neogene pattern aligns with phylogenetic [34] and fossil data [11]. However, the long-term persistence of the LDG from the Cretaceous remains debated. Several studies reported flattened diversity gradients or even richness peaks at higher latitudes, before or after the K-Pg boundary in both terrestrial [12,13] and marine organisms [41]. These studies suggest that the modern LDG may not have been persistent from the Cretaceous onward. Nevertheless, direct comparisons of our simulations outputs wit empirical data should be made cautiously, as our models are specifically tuned for terrestrial fauna.

Our models offer a plausible explanation for this discrepancy. Notably, our modeled *pseudo K-Pg transition*, focusing on shifting climatic gradients, yields a strong tropical peak with elevated speciation and low extinction rates in equatorial regions, contrasting with high extinction and low speciation rates at higher latitudes. This steepens the post-transition LDG, driven by harsher climatic conditions at higher latitudes, increasing ecological filtering in non-tropical regions. In contrast, the K-Pg boundary described in the literature [42] may have temporarily weakened or even reversed the LDG, challenging the idea of its persistence since the Cretaceous. These discrepancies may reflect differences in boundary conditions, particularly climatic, between empirical records and our modeled scenarios. They may also stem from observational biases in terrestrial environments caused by the piecemeal record due to the uneven distribution of fossil localities, along with variations in fossil abundance and preservation.

More information can be derived from diversification dynamics, where our models reveal significant shifts in turnover rates around 35 Ma (Early Oligocene), followed by a steady increase throughout the Neogene. This global signal aligns with findings from several studies [43,44], suggesting that many taxa diversified recently and shaped present biodiversity patterns. High turnover rates have enabled ecosystems to adapt and restructure in response to shifting climates, tectonic changes, and dynamic landscapes. The Early Oligocene also marks the transition into icehouse climate, with steepened temperature gradients between

the equator and poles, which likely amplified environmental heterogeneity and influenced diversification [11]. Further support comes from studies reporting intensified diversification shortly after the Early Oligocene, proposing that the emergence of much of the modern fauna was facilitated by more favorable tectonic or climatic conditions, or a combination of both [43–46].

Taken together, our findings and empirical based studies converge on a key point : the formation of the LDG and the broader question of modern biodiversity origins are tightly linked to tectonics and climate dynamics. In both cases, the emergence of geographic barriers— whether mountain uplift, seaways, or habitat fragmentation—has influenced speciation by restricting dispersal and creating opportunities for allopatric divergence. While the specific mechanisms may differ, these parallels suggest that large-scale environmental changes have played a fundamental role in structuring biodiversity across taxonomic groups.

## Limitations

Due to technical and methodological restrictions, our analysis includes a number of assumptions and simplifications, some of which are described below.

Temporal resolution in mechanistic eco-evolutionary models may influence ecological processes, particularly dispersal dynamics, and shape global biodiversity patterns. In particular, it could affect the modeling of abrupt events such as the K-Pg boundary and their impact on evolutionary trajectories [42,47,48]. However a previous study demonstrated that the age-diversity relationship alone cannot explain global biodiversity patterns, regardless of climatic zone, landmasses or taxonomic group [49], conversely suggesting that temporal resolution may not substantially alter the overall results. Instead, temporal effects are likely to have a greater influence on regional differences in species richness, as certain taxa, particularly those with narrower geographic ranges or shorter evolutionary timescales, may be more affected by these limitations [49].

Additionally, uncertainties in paleoclimatic reconstructions [50] and potential biases in plate model choice [51] must also be considered to ensure robust inferences about deep-time biodiversity dynamics. Future research could benefit from integrating more accurate paleo-precipitation models or downscaling techniques to refine regional predictions. Additional biodiversity data, albeit beyond the scope of the current study would further enhance the reliability of our findings.

## Conclusion

Deciphering current biodiversity patterns from observations is already challenging. Inducing deep time processes driving species diversification and coexistence from the sole use of current, fossil or phylogenetic observations is at this stage possibly unrealistic, given the observational biases. Our model-based, deductive approach permits to circumvent some of these issues, albeit generating others. Our study reveals that physiographic diversity can shape diversification locally or within specific biogeographic regions, without contributing to broad LDG patterns, and highlights that this effect is scale-dependent.

We also emphasize the interplay of abiotic and biotic forces in shaping biodiversity: stable climates promote species persistence and divergence; large landmasses enhance ecological opportunity and facilitate wide dispersal, while the combination of both landscape connectivity and mountainous regions foster isolation-driven speciation, especially at mid- to high latitudes, where climate instability and harshness limit species diversification. Importantly, we reveal in this study the importance of spatial and heterogeneity of landmasses—particularly

the North-South hemispherical asymmetry—in driving large-scale diversity patterns, an often overlooked aspect in LDG studies.

By using mechanistic models compared with fossil and phylogenetic data, this study provides an integrated view of how both biotic and abiotic factors have collectively shaped the LDG, offering a comprehensive framework for understanding the origins of the current large scale biodiversity gradient, which likely persisted since the Cretaceous, steepening and narrowing around 50 Ma, with modern latitudinal patterns of species diversity emerging around 35 million years ago. Together with the datasets made available for physiography and biodiversity, such models provide a basis to appraise the processes species diversification over deep time. Importantly, the persistence of the LDG over deep time highlights the resilience and evolutionary importance of tropical ecosystems. Zooming out of the Anthropocene conservationist perspective, this study consequently suggests that preserving tropical ecosystems is vital not only for its biodiversity but also for safeguarding their role as cradles of species diversification and museums of species persistence. Understanding the stability of tropical ecosystems becomes a priority given their proven critical role in global biodiversity, especially in the face of environmental changes.

## Supporting information

**S1 Fig. Modeled M0 biodiversification across paleolatitudes.** Paleolatitude figures are represented using model M0. Each variable is measured as area-scaled, representing normalized to their maximum mean biodiversity metric per latitudinal degree.
(TIFF)

**S2 Fig. Modeled M1d biodiversification across paleolatitudes.** Paleolatitude figures are represented using model M1d. Each variable is measured as area-scaled, representing normalized to their maximum mean biodiversity metric per latitudinal degree.
(TIFF)

**S3 Fig. Modeled M1s biodiversification across paleolatitudes.** Paleolatitude figures are represented using model M1d. Each variable is measured as area-scaled, representing normalized to their maximum mean biodiversity metric per latitudinal degree.
(TIFF)

**S4 Fig. Schematic LDG dynamics and drivers.** (a) Schematic evolution of the LDG over time, with a significant increase of this pattern from the Eocene onward. (b) Mechanisms driving the LDG: the tropics are not only a cradle of species but also a museum, preserving biodiversity over deep time; driver from abiotic factors, such as climate stability, continental surfaces and connectivity and finally mountain dynamics. Reprinted from Scotese et al. (2018) under a CC BY license, with permission from C.R. Scotese, original copyright 2018.
(TIFF)

## Acknowledgments

The authors acknowledge the Sydney Informatics Hub and the use of the University of Sydney's high performance computing cluster, Artemis, which enabled the execution of $4 \times 100$ eco-evolutionary simulations spanning 150 Ma. Depending on the parameter configuration, each simulation required between 24 and 144 hours of CPU time.

## Author contributions

**Conceptualization:** Manon Lorcery, Laurent Husson, Tristan Salles.

**Formal analysis:** Manon Lorcery.

**Funding acquisition:** Laurent Husson.

**Investigation:** Manon Lorcery.

**Methodology:** Manon Lorcery, Tristan Salles, Oskar Hagen, Alexander Skeels.

**Project administration:** Laurent Husson.

**Supervision:** Laurent Husson, Tristan Salles, Sébastien Lavergne.

**Validation:** Laurent Husson, Tristan Salles, Sébastien Lavergne.

**Visualization:** Manon Lorcery.

**Writing – original draft:** Manon Lorcery.

**Writing – review & editing:** Laurent Husson, Tristan Salles, Sébastien Lavergne, Oskar Hagen, Alexander Skeels.

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
