## [Decision Letter · Decision Letter 0]

29 May 2025

PONE-D-25-21653Deep time evolution of the Latitudinal Diversity Gradient: insights from mechanistic modelsPLOS ONE

Dear Dr. Lorcery,

Thank you for submitting your manuscript to PLOS ONE. After careful consideration, we feel that it has merit but does not fully meet PLOS ONE’s publication criteria as it currently stands. Therefore, we invite you to submit a revised version of the manuscript that addresses the points raised during the review process. See academic editor's comments below.  Please submit your revised manuscript by Jul 13 2025 11:59PM. If you will need more time than this to complete your revisions, please reply to this message or contact the journal office at plosone@plos.org. Please include the following items when submitting your revised manuscript:

We look forward to receiving your revised manuscript.

Kind regards,

Steffen Kiel, Ph.D.

Academic Editor

PLOS ONE

Journal Requirements:

“This study was funded by Université Grenoble Alpes under grand IRGA (Tectonic reshaping of the biosphere)”

5. Please note that your Data Availability Statement is currently missing the repository name. If your manuscript is accepted for publication, you will be asked to provide these details on a very short timeline. We therefore suggest that you provide this information now, though we will not hold up the peer review process if you are unable.

6. We note that Figure 2, 3 and Figure S4 in your submission contain map/satellite images which may be copyrighted. All PLOS content is published under the Creative Commons Attribution License (CC BY 4.0), which means that the manuscript, images, and Supporting Information files will be freely available online, and any third party is permitted to access, download, copy, distribute, and use these materials in any way, even commercially, with proper attribution. For these reasons, we cannot publish previously copyrighted maps or satellite images created using proprietary data, such as Google software (Google Maps, Street View, and Earth). For more information, see our copyright guidelines: http://journals.plos.org/plosone/s/licenses-and-copyright.

 1. You may seek permission from the original copyright holder of Figure 2, 3 and Figure S4 to publish the content specifically under the CC BY 4.0 license. 

Additional Editor Comments:

Dear Manon,

I have received two extensive reviews, both suggesting major revisions, and commenting that the manuscript was (a) too long and wordy, (b) rather unstructured and unfocussed, resulting in (c) rather unclear conclusions, and (d) would clearly benefit from additional and more focused analyses.

Please consider the reviewers’ comments carefully if you decide to submit a revised version.

Kind regards,

Steffen

Reviewers' comments:

Reviewer's Responses to Questions

**Comments to the Author**

1. Is the manuscript technically sound, and do the data support the conclusions?

Reviewer #1: Partly

Reviewer #2: Yes

2. Has the statistical analysis been performed appropriately and rigorously? 

Reviewer #1: I Don't Know

Reviewer #2: Yes

3. Have the authors made all data underlying the findings in their manuscript fully available?

Reviewer #1: No

Reviewer #2: Yes

4. Is the manuscript presented in an intelligible fashion and written in standard English?

Reviewer #1: Yes

Reviewer #2: Yes

5. Review Comments to the Author

Reviewer #1: The origins and deep-time variability of the latitudinal diversity gradient (LDG) have been of immense interest to scientists for centuries, for good reason. I’m admittedly not an expert on eco-evolutionary models like this, but to me, at least, the application of such models to research on the LDG seems interesting and important. Overall I enjoyed reading this paper.

My main comment is that, despite the paper being very, very long (and me reading it very carefully), I still came away completely unsure about what the main results were. A model like this seems like a powerful way to try to tease apart some of the potential mechanisms behind the LDG, but it wasn’t exactly used that way. I think it would help a lot to restructure the paper so that the model is explicitly used for hypothesis testing. For example, take the main potential drivers of the LDG (e.g., energy/productivity, geographical area, biotic interactions, climate stability) and construct a series of models (hypotheses) that test these separately. Using your model, could you reconstruct a modern-day LDG using ONLY energy/productivity? What about ONLY climate stability? Then same for the other factors. Then consider models (hypotheses) with multiple factors. Does a model with energy/productivity AND climate stability perform better than a model with just one of those factors? I think that reframing the paper along these lines would greatly increase its readability and impact.

Secondly, I think the authors need to be very careful about what constitutes a “mechanism” of the LDG. They talk a lot about “climate”, but that’s not an explicit mechanism. They also talk a lot about “speciation rates”, “extinction rates”, and “turnover rates”, but those aren’t explicit mechanisms either. “Available energy” is mentioned in this paper and often in the literature, but what’s the explicit mechanism by which energy would influence diversity? A mechanism would be something like “temperature-dependent speciation” (which Schluter and Pennell [doi.org/10.1038/nature22897] have shown to actually not work empirically as an LDG mechanism), or “extinction rates as a function of temperature variability”, or “available energy reduces extinction rates via the More Individuals Hypothesis”. Determining a clear set of mechanisms from the literature and then “testing” them in silico, would make the paper a lot stronger.

L8-14: The authors argue that there are 3 “primary” drivers of the LDG and a suite of potential “additional” drivers. But at least 2 of the primary drivers have been shown to be unlikely, while several of the “additional” factors (primarily energy and area) are likely the principle factors underlying the LDG. See (1) doi.org/10.1086/508635 and (2) DOI: 10.1016/j.tree.2022.07.013.

L17-19: Rather than being “restricted to extreme and variable greenhouse climates”, flattened LDGs are actually very common in deep time. See Mannion et al. TREE (which you already cite).

L32-34: The contention that “The present-day LDG may have only formed, or at least steepened, in the last 30 to 40 My…” is not supported. There is evidence for tropical-peak LDGs even at certain times in the Paleozoic. Again, see Mannion et al. TREE (2014; which you cite), and Mannion et al. TREE (2022; DOI: 10.1016/j.tree.2022.07.013).

L41: The contention that “[marine] present-day-like LDGs are identified as far back as 252 million years ago” is not fully true. Sun et al. (DOI: 10.1126/science.1224126) found mid-latitude peaks in marine diversity during the Triassic.

Paragraph starting L66: The idea that “physiographic diversity” could play a role in the LDG is interesting, though I must admit I’m a bit confused by the hypothesis. Surely if physiographic diversity was to play a role in the LDG, then such diversity would have to be higher in the tropics. But this seems a hard pill to swallow even abiotically, but leaving aside the substantial evidence (e.g. doi:10.1038/nature14949) that even on high tropical mountains, many of the taxa come from higher latitudes rather than from the nearby tropical lowlands.

L95: For the precipitation data, the authors cite Valdes et al. (doi.org/10.5194/cp-17-1483-2021), but this might not be the correct citation, as Valdes et al. don’t provide paleo-precipitation reconstructions.

L149 “Ecology” Section: I wasn’t clear from the description if local population size influences speciation rate, but it certainly seems like it should.

Methods section: I’m not familiar with the gen3sis model so I can’t adequately review this part of the paper. It would be useful for the authors to provide even just a little bit more broad overview of this model. Is it run via a Python or R package? Is it an inscrutable FORTRAN code like a GCM? Have prior applications of this model tended to produce empirically-verifiable results?

Methods section: How is geographical area incorporated into your model? Prominent hypotheses posit that the tropics have greater diversity because, integrated over time, megathermal landmasses have been bigger. It seems like a species-area relationship should somehow be implemented into the model.

Methods section: I couldn’t find any information on where the present-day mammal data came from that you used to validate your model.

L281: consider replacing “extrude” with “extend” or “extrapolate”

L318: replace “predominantly located” with “particularly high”

L328: “complementary” to what?

L341-342: what exactly do you mean by “narrow” in this sense?

L342 and onward: why is the K-Pg transition referred to as “pseudo”?

Section starting L431: In the Intro, the authors presented a series of hypotheses from the literature about why there is (and often, but not always, has been) an LDG. But now in this section, none of those drivers are assessed, just “plate tectonics”.

Results section: This is very long and therefore a bit difficult to follow. Could it be condensed and/or could some of the material be moved to a Supplementary Material section?

Discussion section: Also very long!

L482-485: Now another set of potential drivers of the LDG is presented, different from those discussed in the Intro and the Results.

L482-485: It seems like you might be conflating the “area” hypothesis with the “habitat heterogeneity hypothesis”. While it’s true that habitat heterogeneity is often used as a mechanism to explain species-area relationships (SARs), SARs can also occur even with no habitat heterogeneity at all. See, for example, the “More Individuals Hypothesis” from Lawton, I believe.

The figure captions are embedded in the manuscript, but the figures themselves are all at the end. It would help a lot to have each caption next to its associated figure.

Reviewer #2: This manuscript presents a simulation-based investigation into the origins and temporal dynamics of the Latitudinal Diversity Gradient (LDG) in terrestrial mammals. Using mechanistic models informed by fossil and phylogenetic data, the authors simulate biodiversity patterns from the Cretaceous to the present. The models reveal the emergence and steepening of the LDG beginning around the Late Cretaceous to the early Paleogene (~50–35 Ma), shaped by tropical speciation, poleward dispersal, extinction gradients, and geographic asymmetries. The findings are generally consistent with the "Out of the Tropics" hypothesis, and emphasize the roles of both abiotic and biotic drivers in shaping global biodiversity patterns.

The study is highly ambitious in scope and tackles a long-standing question in macroecology: the emergence and persistence of the LDG. The integration of simulations with fossil and phylogenetic evidence is commendable, as is the emphasis on deep time dynamics. However, several points require clarification or expansion to strengthen the manuscript’s impact and clarity:

- The paper lacks a clear conceptual framework. The introduction could better define key terms (e.g., LDG, diversification vs. speciation), summarize aims more explicitly, and structure the various hypotheses and drivers in a more organized way (e.g., abiotic vs. biotic, temporal vs. spatial). The current structure feels list-like and sometimes inconsistent between sections (e.g., intro vs. discussion).

- While the modeling framework is rich, the paper remains overly descriptive in presenting results and does not fully exploit the model’s potential to test mechanisms or disentangle the roles of different processes. Several general and specific comments request deeper comparative analysis between model scenarios and more explicit discussion of the model’s assumptions, simplifications (e.g., speciation), and limitations.

- The writing is often dense, technical, or ambiguous, especially in methodological sections. Key processes (e.g., dispersal, turnover, ecological dynamics) are introduced with vague or overly complex language, and some terms (e.g., “surface processes”, “cost function”) are used without clarification. Readers unfamiliar with grid-based simulation models may struggle to follow the logic without additional explanations, definitions, or simplification.

- The paper sometimes fails to tie together its aims, methods, and interpretations. For example, some ideas appear in the discussion without being developed earlier, and the conclusion repeats previous points without offering a strong synthesis. The discussion does not fully explain why certain patterns (e.g., turnover asymmetry, cradle/museum dynamics) arise, and conservation implications remain underdeveloped despite being potentially important.

Below you will find more general and specific comments associated with specific line numbers in the ms.

General comments

1. 8-11: The reader may want to know how these biotic factors relate or differ from common ideas about "time for speciation", "speciation rates", and "ecological limits".

2. 13-19: Same general comment as above, i.e. can these multiple factors be organized in some way that makes it less like a list and more like a categorization of different temporal, ecological, and evolutionary effects?

3. 22-26: Here the authors seem to try to connect the list of factors above with specific hypotheses that are based on speciation, migration, and diversification. However, the link is not clear. A better structuring of the text and maybe even a conceptual figure would help.

4. 67-76: It seems like the authors are trying to specify their modeling scope in this last paragraph but it is not 100% clear. I suggest being even more clear on exactly what processes and mechanisms are modeled and why. Also, they may consider merging this paragraph with the preceding one as I think it would improve the flow of the introduction.

5. 111-115: Here the authors probably want to do a better job describing exactly what the biodiversity processes and mechanisms are. Maybe a conceptual figure or a table would help.

6. 117: It is unclear and a bit confusing to the reader, as it is not clear if these model specifics are included in all the scenarios mentioned above (also the ones that are "solely based on climate and tectonics").

7. 121-130: The way dispersal is framed as some cost is not intuitive to me as a first-time reader. I was hoping that the authors can frame this in a more intuitive way with more focus on actual dispersal and less on costs.

8. 132-136: I am sure the authors realize that this is an extremely simplified view of speciation. This is of course fine in a modelling study but the authors need to discuss the potential consequences of such simplifications for the results.

9. 138-147: This section is described as if it was a Ornstin-Uhlenbeck process but it is really not as OU processes are commonly described as a continuous model. At the same time the authors mix in concepts like Brownian motion which is commonly completely undirected. This is confusing. I suggest that the authors revise this part such that it goes more in line with the standard terminology of directed stochastic models.

10. 226-239: For someone that is not an expert in these types of grid-based models this section becomes very dense. I suggest the authors try to revise this such that it becomes more accessible to non-experts.

11. 253-261: Seems more appropriate to present this in the methods section. A reminder here in the results may be justified but should also be clearly described in the methods section.

12. 272-276: I may have missed somthing but I think that the authors fall short in actually "Understanding the LDG". Rather than concluding that biodiversity is correlated with precipitation (which is a trivial observation) they can actually disentangle some of the more interesting explanations that are based on dispersal, ecology, adaptation, and speciation. I thus call for a more indepth analyses of the different model scenarios Table 1 and how they preform and why.

13. 293: General: This whole section is very descriptive with very little explanations of why these patterns emerge. Given that the authors run different models (Table 1) they should be able to provide insights to the underpinnings of the observed patterns.

14. 376-377: This may be the most concerning result that comes with this study — the fact that different model scenarios give very similar outputs. It essentially means that the authors have limited information from the models that can help them disentangle the relative importance of different factors affecting the biodiversity patterns. The major goal of this paper thus seems to be out of scope for the particular model and model scenarios that were chosen by the authors.

15. 431: In this section the authors provide some detail on potential drivers of biodiversity, but they are all more or less abiotic or environmental effects. Such explanations are quite different from the focus that was introduced in the introduction and early parts of the model presentation, i.e. dispersal, ecology, adaptation, and speciation. Thus it seems to me like the authors are missing their own mark that was set up in the introduction and early parts of the model description.

16. 522: Quite wordy section that does not provide a lot of insight. Repeating some of the key results and telling the reader that similar (or dissimilar)...

17. 568: Although the discussion about cradle and museum is interesting, I cannot help but ask the "why" question. I was hoping for more answers along the lines of the questions asked in the introduction of this paper.

18. 621: In this section the authors are scratching the surface of what I am calling for above. I suggest the authors try to have such analyses of the comparison between model scenarios permeate the paper much better. This is where I see the real contribution from this modeling study.

19. 699: This conclusions section does not bring much new or outstanding information. I suspect that this whole section can be deleted without loss of readability or clarity — given that the other parts of the discussion are tightened, that is.

Specific Comments

1. Watch out for long sentences in abstract. Also, the abstract could benefit from ending with an outlook and future perspectives sentence.

2. 21-22: Definitions and distinctions between diversification and speciation probably needed here.

3. 49-50: A few follow-up sentences that provide explicit examples of mechanisms that have been modeled, and specific modeling studies and their results, may be good to add here, as it would help the reader understand what is meant and what possibilities these types of models provide.

4. 65: Even though the authors list key features of the model above, it may be worth being very clear — i.e., reminding the reader about what mechanisms they are focused on in this study.

5. 81: “These dynamics”, unclear — be specific.

6. 86-87: Not sure what this sentence is trying to say. I suggest either clarifying how the Game of Life model has inspired Gen3sis or deleting the sentence altogether.

7. 93: Not sure what this means — please clarify.

8. 93: The reader is kept in the dark about what these biological functions and behavioral laws may be. I suggest that the authors provide a few examples here to relieve the reader’s frustration a bit.

9. 121: Please specify the temporal extent of such a time step.

10. 121: Terminology used here implies that the reader should know what a "cost function" is and how it connects to dispersal. This is, however, not clear to me.

11. 132-136: As I am sure the authors realize, this is an extremely simplified view of speciation. This is of course fine in a modeling study like this, but the authors need to discuss the potential consequences of such simplifications for the results.

12. 138-147: This section is described as if it were an Ornstein-Uhlenbeck process, but it is really not, as OU processes are commonly described as continuous models. At the same time, the authors mix in concepts like Brownian motion, which is commonly completely undirected. This is confusing. I suggest that the authors revise this part such that it aligns more closely with standard terminology for directed stochastic models.

13. 176-177: It may be worth describing why such an approach is conducted — i.e., a follow-up sentence describing how these 100 simulations were treated in downstream analyses.

14. 179: These 1 Myr time steps are huge in the context of dispersal and ecology. I strongly suggest that the authors discuss what consequences such long time steps, combined with the fact that different modeled processes act on very different time scales, have for the model behavior and output.

15. 194-194: Should this not be part of the model specifics on dispersal presented above?

16. 199: See comment above about OU process.

17. 221-224: Seems more like a discussion point than a statement in the methods section.

18. Figure 4 would benefit from panel titles telling the reader which panel is showing which hemisphere results.

19. 282: The authors fall short in justifying this claim. See my general call for in-depth analyses of the different model scenarios and how they perform — and why.

20. 284-285: Not sure this is suitable for the results section — rather part of a discussion.

21. 519-520: Yes, I agree that this is what the model can do and should do. But unfortunately, I do not think the authors utilize their model to its full potential. Rather than explaining the mechanisms underpinning LDGs, they resort to a very descriptive presentation of their results that essentially boils down to environmental and/or abiotic factors — despite the fact that they frame their study from several biotic perspectives in the introduction.

6. PLOS authors have the option to publish the peer review history of their article (what does this mean?). If published, this will include your full peer review and any attached files.

Reviewer #1: No

Reviewer #2: No

---

## [Author Response · Author response to Decision Letter 1]

24 Jul 2025

Dear Reviewers,

We would like to thank you sincerely for your time and effort in reviewing our manuscript. We appreciate your constructive comments and are pleased that you found our study interesting and potentially suitable for publication in PLoS One.

We have carefully considered all your suggestions and have substantially revised the manuscript in response. We provide a point-by-point response to each of your comments in the « Response to Reviewers » document, indicating the changes made and clarifying aspects of the study where necessary.

In summary, the major revisions include:

1. Restructuring the manuscript: We revised the overall structure to streamline the narrative, removing the focus on a narrow set of hypotheses and focusing more broadly on the fundamental ways paleoenvironmental change shape biodiversity gradients impacting speciation, extinction and dispersal.

2. Expanded methodological detail: We have included more information on the modeling approach to enhance transparency and reproducibility, as well as to improve the reader’s understanding of the simulation framework.

3. Concise results and discussion: We significantly shortened and clarified the results and discussion sections by reducing wordy descriptions, focusing better on the key findings.

In addition to the revised manuscript, we have provided a tracked-changes version in PDF format that highlights all modifications made since the initial submission.

We hope these revisions adequately address your concerns and bring the manuscript to the standard expected for publication in PLoS One. Thank you again for your valuable feedback and thoughtful recommendations.

Sincerely,

Manon Lorcery and co-authors

---

## [Editor Report · Decision Letter 1]

1 Sep 2025

PONE-D-25-21653R1Deep time evolution of the Latitudinal Diversity Gradient: insights from mechanistic modelsPLOS ONE

Dear Dr. Lorcery,

Thank you for submitting your manuscript to PLOS ONE. After careful consideration, we feel that it has merit but does not fully meet PLOS ONE’s publication criteria as it currently stands. Therefore, we invite you to submit a revised version of the manuscript that addresses the points raised during the review process. See my comments below. Please submit your revised manuscript by Oct 16 2025 11:59PM. If you will need more time than this to complete your revisions, please reply to this message or contact the journal office at plosone@plos.org. Please include the following items when submitting your revised manuscript:

We look forward to receiving your revised manuscript.

Kind regards,

Steffen Kiel, Ph.D.

Academic Editor

PLOS ONE

Journal Requirements:

Additional Editor Comments:

Apologies for the delay – your manuscript arrived the day after I left for my summer holidays.

I carefully read your response to the reviews and the revised manuscript. Below I outline a few issues that need fixing or addressing, but once that’s done, I’m happy to accept your manuscript.

Steffen Kiel

Comments, mostly by line number:

abstract & throughout the manuscript: please use ’Cenozoic’ instead of ‘Tertiary’ – the latter term has been abandoned more than a decade ago.

Line 5, perhaps ’recognized’ instead of documented.

21, ‘has been’

34, permits

Fig. 1 caption: remove the last sentence, it just repeats the main text.

87-88, please be clearer about M1s and M1d

440, amplified

463, insert comma after ‘stability’

478, insert comma after ‘specialization’

498-500, well, you did not investigate the time before the Cretaceous, so you cannot really point to a Cretaceous origin of the LDG.

520-529, perhaps consider that the world entered an ice-house climate in the early Oligocene, with steepened temperature gradients from equator to poles.

---

## [Author Response · Author response to Decision Letter 2]

2 Sep 2025

Dear Dr Kiel,

We would like to thank you again for your careful evaluation of our manuscript and for forwarding us your helpfull comments. We are pleased that the remaining comments are minor and we have revised the manuscript accordingly.

In particular, we have modified the manuscript based on the following minor comments:

• abstract & throughout the manuscript: please use ’Cenozoic’ instead of ‘Tertiary’ – the latter term has been abandoned more than a decade ago.

That is right, we have modified accordingly (abstract and lines 284 and 317).

• Line 5, perhaps ’recognized’ instead of documented.

We replaced « documented » by « recognized » line 5.

• 21, ‘has been’

We corrected « have been » by « has been » line 21.

• 34, permits

We corrected « permit » by « permits » line 34.

• Fig. 1 caption: remove the last sentence, it just repeats the main text.

Great comment, we removed the last sentence of the Fig. 1 caption.

• 87-88, please be clearer about M1s and M1d

We clarified the differences between M1s and M1d as you pointed out, it was not very clear. To do so we stated : « Second scenario M1s integrates physical barriers (Φ) into speciation, with dispersal based on geographic distances (∆). Dispersal in M1d is based on both geographic distances and physical barriers (∆ + Φ), with speciation depending solely on geographic distances (∆). » (Lines 87-90).

• 440, amplified

We corrected « amplify » by « amplified » line 442.

• 463, insert comma after ‘stability’

We added a comma after « stability » line 465.

• 478, insert comma after ‘specialization’

We added a comma after « specialization » line 480.

• 498-500, well, you did not investigate the time before the Cretaceous, so you cannot really point to a Cretaceous origin of the LDG.

That is true, good point. To clarify this we now state « The main trend that our models show is that the LDG became established during the Cretaceous, and that the present-day LDG did so at around 50 Ma, then steepening, and taking its current form by approximately 35 Ma. » (Lines 500-502)

• 520-529, perhaps consider that the world entered an ice-house climate in the early Oligocene, with steepened temperature gradients from equator to poles.

Great remark, we have added the idea that the Early Oligocene also marks the transition into icehouse climate, with steepened temperature gradients between the equator and poles, likely amplifying environmental heterogeneity and influencing diversification (Lines 522-533).

We believe these revisions fully address the minor comments and further improve the clarity and precision of the manuscript. We are grateful for the constructive feedback and for the opportunity to revise our work.

Thank you very much for your time and consideration.

Sincerely,

Manon Lorcery and co-authors

---

## [Editor Report · Decision Letter 2]

4 Sep 2025

Deep time evolution of the Latitudinal Diversity Gradient: insights from mechanistic models

PONE-D-25-21653R2

Dear Dr. Lorcery,

We’re pleased to inform you that your manuscript has been judged scientifically suitable for publication and will be formally accepted for publication once it meets all outstanding technical requirements.

Kind regards,

Steffen Kiel, Ph.D.

Academic Editor

PLOS ONE